# All Simulations Are Not Equal: Simulation Reweighing for Imperfect Information Games

## Abstract

Imperfect information games are challenging benchmarks for artificial intelligent systems. To reason and plan under uncertainty is a key towards general AI. Traditionally, large amounts of simulations are used in imperfect information games, and they sometimes perform sub-optimally due to large state and action spaces. In this work, we propose a simulation reweighing mechanism using neural networks. It performs backwards verification to public previous actions and assign proper belief weights to the simulations from the information set of the current observation, using an incomplete state solver network (ISSN). We use simulation reweighing in the playing phase of the game contract bridge, and show that it outperforms previous state-of-the-art Monte Carlo simulation based methods, and achieves better play per decision.

## 1 Introduction

Board games like Chess and Go (Campbell et al., 2002; Silver et al., 2018) served as a long-standing benchmark for AI, and they witnessed the advancement of (deep) reinforcement learning. Past successes in these benchmarks, however, are mostly about perfect information games. In the real world, agents will never have access to all ground truth information and have to make decision under missing information. Thus, theories and strategies learned from imperfect information games can be better transferred to these environments. It was not until recently researchers have focuses more on imperfect information games (Heinrich & Silver, 2016; Brown & Sandholm, 2018; 2019; Foerster et al., 2018b).

One key challenge in incomplete information games is to model beliefs. In perfect information games like Go and Chess, the best action only depends on the current state. Simulation, reasoning and planning is relatively easy because it is independent of the opponent and history actions, and all information is public. On the other hand, for imperfect information games like Poker (Brown & Sandholm, 2018; 2019) and Contract Bridge, an agent needs to infer which state it is from the information set of the observation, which varies dramatically on different opponents and history actions. The agent also needs to model actions of other agents, which might be collaborative or adversarial.

There are multiple ways to address this problem. Early approaches use heuristic based system to assign different priors to possible states. Simulation based approach samples possible states and try to convert the problem into a perfect information one. Each state is solved independently and accumulated together to determine the best action. An example is contract bridge bot GIB (Ginsberg, 1999). Prior works have tried to model belief explicitly with neural networks, and try to optimize it with reconstruction loss (Rong et al., 2019). Recently researchers have also tried to model belief implicitly in an end-to-end manner to capture all information given the history actions.

However, these approaches all have various shortcomings. Rule-based systems are restricted in their representation power. Simulation based approaches usually require a large number of simulations to get to relatively accurate results, and are very computationally expensive. While end-to-end methods can give good results, its learned latent space is hard to interpret.

In this paper, we would like to model belief in a computationally efficient manner, while still keeping the interpretability. For this, we propose to backtrack from the current observation using previous action sequences of the existing player and opponent. Given the current observation, we first sample

a few underlying complete information states. For each sampled state, we rewind the history action to reconstruct all the history states. Then at each time step we measure how likely the agent have took that action, rather than the other actions, if this sampled history state is the ground truth. The posterior probability is calculated for each sample. At last, the final belief consists of weighted samples of the simulations rather than a uniform distribution, and is more accurate.

To evaluate the likelihood for all agents in all past time steps is computationally expensive. This is because at past time steps the other agents also only observe incomplete information. To get an accurate estimate of action likelihood, *another* round of simulation needs to be done, for each of these scenarios, since this is what the other agents use in the past time step. To solve this problem, we propose to use an Incomplete State Solver Network (ISSN) to imitate the results of simulations. We use this network to predict the action likelihood of a trajectory, and assign belief weights to each sample. We show that through simulation reweighing, the result is improved, when the mechanism is applied to the playing phase of contract bridge.

Although we test on Contract Bridge, the proposed technique does not rely on any domain specific knowledge in this process, and is general for any imperfect information games. We will open source all the code and data used to inspire further research in this direction.

## 2 CONTRACT BRIDGE

Contract bridge is a game of 4 players. Two teams of 2 players are competing against each other. It is played with one full deck of cards, and each player is dealt with 13 cards. The game has two phases, bidding and playing.

### 2.1 BIDDING PHASE

During the bidding phase 4 players take turns to suggest a target contract, which promises the number of winning rounds in the playing phase as well as the trump suit. The bids cannot decrease in nominal value. If a player is not willing to bid a higher contract, the `Pass` bid is always available. There are also some special bids, namely `Double` and `Redouble`, which can be used to adjust the scoring of the same contract. The bidding phase ends when no one is willing to suggest a high contract than the current one. During the bidding phase, players usually convey extra information to their partner through the bids themselves, in order to reach the optimal contract.

### 2.2 PLAYING PHASE

After the bidding phase a target contract is determined. The target contract specifies the trump suit and the number of tricks needed for the declaring side. For the playing phase, the partnership who wins the contract is the declaring side and the other partnership is the defending side. The player who first bids the suit of the target contract in the declaring side is the declarer. There are 13 rounds (called tricks) in the playing phase and the declaring side tries to accomplish the target contract.

During the playing phase, the defending side plays first (called `lead`). After the lead, the partner of the declarer immediately lays his hand down (called dummy), and it is visible to everyone. The declarer controls both his play and dummy's play. There is a total of 13 rounds, and each round every player can only play one card. The players are required to follow suit. If they are out of the led suit, a trump suit card can beat it, while discarding non-trump suit will be losing the trick. The rank of the cards from largest to smallest is A, K, ..., 3, 2. The player with the best card wins the current trick, and plays first next round. Once the playing phase finishes, tricks won by each side are accumulated. If declarer wins enough tricks to reach the target contract, he `makes` the contract, otherwise the contract is `down`. Scores are calculated according to the target contract and the tricks taken.

## 3 RELATED WORK

Recently there are quite a few well-known works on imperfect information games. Card games are most popular among these. Texas Holdem solvers focus on finding the Nash Equilibrium through

counterfactual regret minimization (Zinkevich et al., 2008). Libratus (Brown & Sandholm, 2018) utilizes nested safe subgame solving to reach very low exploitability. It can also enhance the background blueprint strategy in real time using a self-improver. DeepStack (Moravčík et al., 2017) uses a value network to approximate value function. They both outperform top human professional players. Bayesian Action Decoder (BAD)(Foerster et al., 2018b) proposes to model public belief and private belief separately, and sample policy based on an evolving deterministic communication protocol. This protocol is then improved through Bayesian updates. BAD is able to reach near optimal results in two-player Hanabi, outperforming previous methods by a significant margin.

The contract bridge game itself has also attracted multiple researcher's attention. There are multiple works trying to solve automatic contract bridge bidding (Yeh & Lin, 2016; Rong et al., 2019). There are also commercial software programs which uses a combination of heuristic and simulation methods to make superior moves during the playing phase of contract bridge. Among those GIB(Ginsberg, 1999), Jack[1] and Wbridge5 [2] are the most notable ones. They are championship winning programs in world computer bridge tournament. GIB won in 1998 and 1999, whereas from 2001 to 2018 Jack and Wbridge5 snatched 16 titles out of 18.

Game theoretical methods (Myerson, 2013; Dudik & Gordon, 2009) are principled ways to solve incomplete information games. However, they are computationally prohibitive with large states (Sandholm, 2010). Simulation methods are crucial in game solving, for both perfect information and imperfect information games. MCMC algorithms are proposed and studied (Roberts & Rosenthal, 2004) . Strong agents in perfect information games (Campbell et al., 2002; Silver et al., 2016; 2017; 2018) relies heavily on simulation and/or knowledge distilled from simulations. Recently it is shown that simulations can also be efficiently done in imperfect information games setting (Brown & Sandholm, 2018; Moravčík et al., 2017).

Belief modeling is a key factor in imperfect information games. Monte Carlo Sampling from known information, known as *determination*, is a basic technique to models beliefs given observations (Whitehouse et al., 2011; Ginsberg, 1999; Buro et al., 2009; Ward & Cowling, 2009; Schäfer et al., 2008). MCTS under imperfect information has also been studied in Whitehouse (2014). Since brute-force sampling in large (potentially exponential) information set is inefficient, some prior works on card games models belief via independent assumptions (Foerster et al., 2018b) or via latent neural representations (Rong et al., 2019; Tian et al., 2018). Besides them, LOLA (Foerster et al., 2018a) agents learn with anticipated learning of opponents to reach stronger performance. There are also attempts to model belief in real time strategy games (Synnaeve et al., 2018; Tian & Gong, 2018).

Reweighing, or prioritizing examples in training is a common concept. It is commonly used to speed up training or make training more robust. One example usage in reinforcement learning is prioritized experience replay (Schaul et al., 2015). Researchers have also worked on adaptive rejection sampling methods, such as Gilks & Wild (1992) and more recently Maddison et al. (2014).

# 4 SIMULATION REWEIGHING

## 4.1 PROBLEM SETUP

We prepare a dataset of bridge deals from expert tournament play. It contains 84k deals and 2.4 million states from the playing phase. During some deals, the playing phase finishes prematurely because an agreement is made on the remaining tricks, if the remaining play is obvious. We split the dataset into a 90% training set and a 10% testing set. Double Dummy Solver (DDS) [3] computes the maximum tricks each side can get if all the plays are optimal. We run DDS on all 2.4 million positions, to get the corresponding results for each available action. This serves as ground truth of the current position. For each position, we run 50 simulations to compute the baselines and serve as training target of the pretrained network. Each call to DDS takes about 5 milliseconds.

---

[1]http://www.jackbridge.com/eindex.htm
[2]http://www.wbridge5.com/
[3]https://github.com/dds-bridge/dds

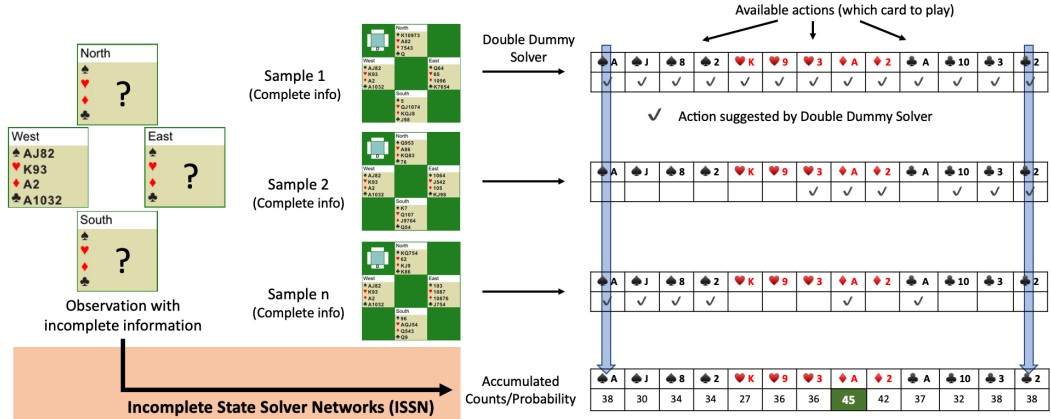

Figure 1: Incomplete State Solver Network (ISSN). Given an observation with incomplete informa-tion, traditional approaches (e.g., GIB) first sample its possible complete information states, solve them to get good actions and accumulate the good actions to get a accumulated count/probability of each action. In contrast, we directly learn a Incomplete State Solver Network (ISSN) to predict which actions are likely given the incomplete information observation.

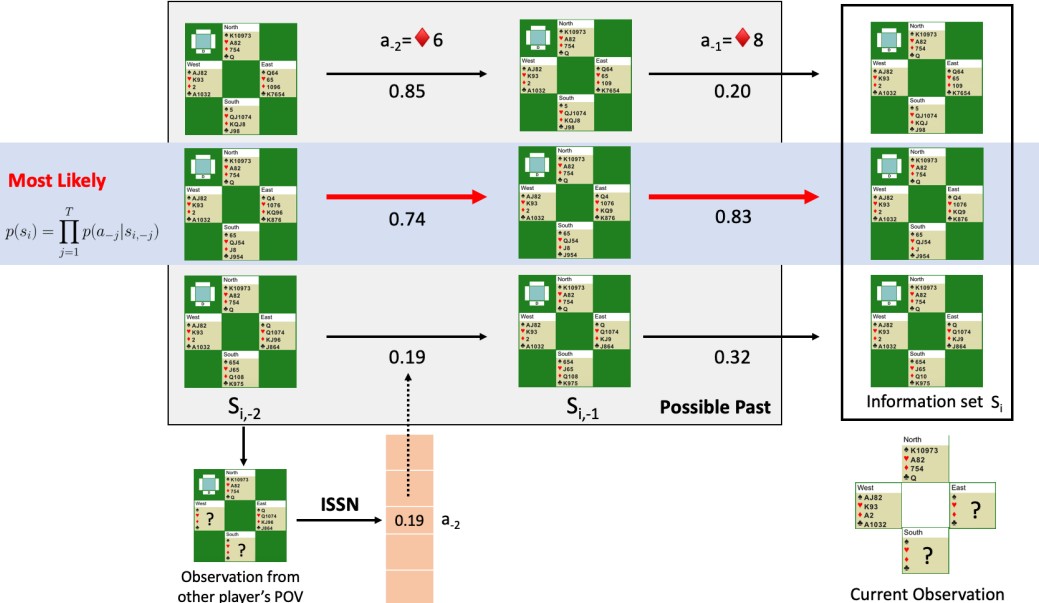

Figure 2: Reweighing sampled states by backtracking. Given the current observation $o_t$, we sample possible complete states $s_t^i$ and check whether they likely happen given the previous actions. This is done via checking whether the likelihood of each previous actions $a_{-j}$ is high given the backtracked state, using ISSN. Note that ISSN takes an incomplete state as the input to mimic how the other players reason about the current situations from what they can see. Disclaimer: the probabilities are not real results from trained network.

## 4.2 GIB BASELINE

Suggested by the GIB paper, for a given position, a Monte Carlo sampling is conducted. A set of k deals is constructed by shuffling the unknown positions and each deal should be consistent with previous history (e.g. if a player has discarded some other suit during a previous spade play, he should not have any more spades left). GIB uses additional human designed heuristics from bidding and playing to check consistency. For simplicity, in this work we just use discarding information to check consistency. For each simulation DDS score is computed for each available move. The move with a maximum expected score during these simulations is selected.

### 4.3 METHOD

The drawback of the baseline is obvious. First, it is mostly independent on the history actions except for the consistent shuffling. The information gained in the previous actions is mostly lost. Second, a DDS run is very slow, since it conducts a full search from the current position. Furthermore, to get relatively accurate results, a large amount of simulations are needed. This results in slow playing.

We propose a new procedure to improve the above process. We pretrain two networks, one with imperfect information and one with perfect information, both to approximate the simulation results of DDS. We call the imperfect information network `Incomplete State Solver Network (ISSN)`, and perfect information network `State Solver Network (SSN)`. For each hand position $k$ simulations of DDS are run. For each simulation, the moves with the optimal number of tricks are marked as the optimal moves. We sum up the optimal moves counter in these k simulations. This results in a counter for each legal move and we treat this as the training target. The process is described in Figure 1.

Once the pretrained network is ready, we use the networks in the following manner: For each position, we still sample k deals. Instead of using DDS calculations for each of these simulations, we use the ISSN to get the optimal move counter for each legal move $d(s_i, a)$. In previous works, optimal moves counter is accumulated across all simulations; In our approach, all simulations are not equal and they have to be reweighed by ISSN. From each simulation of the current positions $s_i$, we check the past T history moves. The history states given the simulation state are reconstructed, noted as $s_{i,-j}$ where $j$ ranges from 1 to $T$. For all $i$ and $j$ we use the Incomplete State Solver Network to calculate the probabilities of actual moves selected $p(a_{-j}|s_{i,-j})$. The posterior probability of simulation $s_i$ is then $\prod_{j=1}^{T} p(a_{-j}|s_{i,-j})$. Finally, the moves suggested by the reweighed simulations is shown in Equation 1.

$$\max_a \sum_i \prod_{j=1}^{T} p(a_{-j}|s_{i,-j})d(s_i, a) \qquad (1)$$

The aforementioned approach is shown in Figure 2.

### 4.4 TRAINING DETAILS

#### 4.4.1 FEATURES

We extract the following features from a state. We use 4x52 bits to represent the remaining cards in all players' hands. There is a proper mask, indicating if the cards are visible to the current player, if we are training with imperfect information. The next 52 bits indicates if the current card is a trump. Then, we use 3x52 bits to represent the cards in play in the current round, since up to 3 players may have played a card. Finally we use an additional 16 bits to indicate the discarding information in history moves. If player i does not follow suit to suit j at some point in the history, the corresponding bit is marked with 1. These features are concatenated together. We use data augmentation to shuffle the suits, and changes the input features and targets correspondingly.

#### 4.4.2 NETWORK

The network uses an initial fully connected layer to transform the input representation, and then 8 fully connected layer and 800 hidden units. A skip connection is added every 2 layers. A last fully connected layer to 52 neurons is added to get the resulting policy. Finally, illegal actions of the policy are masked out and the resulting weights are re-normalized.

The feature representation and network architecture is shown in Figure 3.

#### 4.4.3 TRAINING PROCEDURE

For each position in the dataset, we run k = 50 simulations. We solve each of these simulations using DDS. To train ISSN, for each original deal and simulations, optimal moves counter is then calculated for each legal action. We normalize the counter to make this a probability distribution, and each action's probability is directly proportional to the counter. The loss function is the Kullback-Leibler

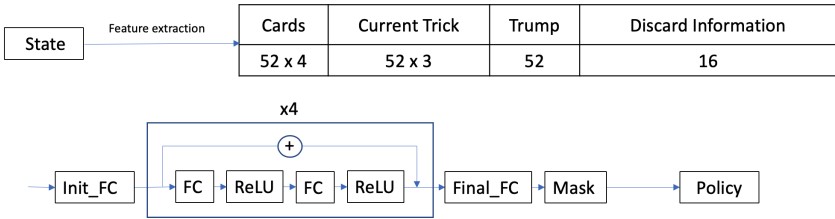

Figure 3: Feature representation and network architecture of ISSN.

Table 1: Pretrained network strength compared with DDS ground truth. Results are recorded in the metric of trick loss per state.

| Network | Cost |
|---|---|
| ISSN | 0.064 |
| SSN | 0.024 |
| DDS | 0 |

divergence between the DDS simulation results and network results. To train SSN, we just use the original deal's DDS results counter as the training target. The simulation results are discarded in this case. We train the model with ADAM optimizer (Kingma & Ba, 2014) with a learning rate of 1e-4.

## 5 EXPERIMENT RESULTS

We set up the evaluation metric as the average tricks loss from optimal play assuming perfect information. The ground truth play is the DDS result for the original deal and all simulation deals. We compare the following algorithm with the ground truth to get the costs of the policy.

### 5.1 PRETRAINED NETWORKS

Pretrained networks are trained to imitate the DDS results. However, due to network representation power and imperfect information, there will be a performance gap. We show the strength of pretrained networks in Table 1.

### 5.2 BASELINES

We implement three baselines. The first baseline is similar to the GIB Baseline approach, where we accumulate all the DDS simulation results, and pick the best action. We denote this baseline as `dds-baseline`. We implement another baseline which also uses the belief reweighing technique, but without using the pretrain networks. To accomplish this, given a hand position, we first run k simulations. For each simulations, we need to reconstruct history states, and check the probability of history actions. Thus for each reconstructed state we need to again run k simulations, consistent with the reconstructed states, to get all the DDS scores, and compute the probability to take the specified move $p(a_{-j}|s_{i,-j})$. We denote this baseline as `DDS + DDS`. Since the Incomplete State Solver Network is an imperfect information network learning from the DDS data to estimate $p(a_{-j}|s_{i,-j})$, `DDS + DDS` serves as an upper bound that ISSN can reach. We should note that `DDS + DDS` runs $k^2$ simulations for each position and is very computationally expensive. The third baseline does not use DDS information but directly predict the action from imperfect information state. We refer to this as `nn-baseline`.

### 5.3 MAIN RESULTS

We compare our approaches with the baseline methods, and compute the cost of the approaches. We pick $T = 3$ in pretrained networks.

First, We examine the performance using the Incomplete State Solver Network in junction with DDS, noted as ISSN + DDS. It outperforms `dds-baseline` using DDS simulations only. The weighted simulations are a better approximate to possible hidden states the agent is in. The performance of ISSN + DDS is also close to `DDS + DDS`, which is the upper bound it can reach.

Table 2: Comparison between neural belief simulator and baselines. Results are recorded in the metric of trick loss per state.

| Method | Simulation Method | Reweighing Method | DDS calls | NN calls | Cost |
|---|---|---|---|---|---|
| dds-baseline | DDS | None | $k$ | 0 | 0.0389 |
| ISSN + DDS | DDS | ISSN | $k$ | $kT$ | 0.0380 |
| DDS + DDS | DDS | DDS | $k^2T$ | 0 | 0.0378 |
| nn-baseline | ISSN | None | 0 | 1 | 0.0640 |
| SSN | SSN | None | 0 | $k$ | 0.0565 |
| ISSN + SSN | SSN | ISSN | 0 | $k(T+1)$ | 0.0554 |

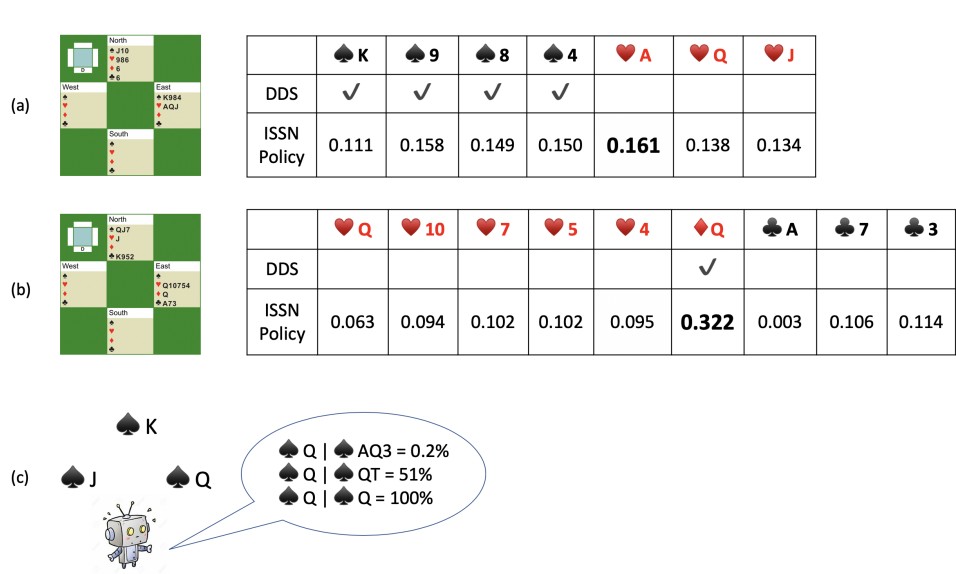

Figure 4: Examples and results using the final model. (a) and (b) is from test data, where in (a) ISSN fails to find the best move, and in (b) ISSN finds the best move. (c) is from simulation data, where the agent is to make a decision on the unusual play of ♠Q.

Second, we examine the performances of approaches without using DDS. These approaches run very fast. The use of State Solver Network with k simulations outperforms the `nn-baseline`. We also check the performance using both Incomplete State Solver Network and State Solver Network, noted as ISSN + SSN. It outperforms using SSN only. This demonstrates that the reweighing technique can also work well with neural network simulation results.

In both approaches, ISSN and simulation reweighing helps to improve the performances. The result is shown in Table 2.

## 6 EXAMPLES

We feed a few examples to our final trained networks to check their performances. The results are shown in Figure 4. (a) and (b) are from test data. In (a) it is a no-trump contract. The agent has to decide which suit to attack. It is a close call and the network suggests to play ♡A, where the optimal play is to play spades. In (b) trumps are diamonds. The agent is able to clearly locate the only best play suggested by DDS ◇Q. (c) is from simulation data to demonstrate that ISSN is trained well. The agent is making a decision where ♠J, ♠K, ♠Q has been played in the current trick. ♠Q is an unusual play under opponent's ♠K; At runtime, the agent simulates k times, and we pick 3 simulations among them. In the first simulation the previous player has ♠AQ3. If that is the case, playing ♠Q from ♠AQ3 is a bad play. if you play A you will win the current trick, and since AK are already played, Q is very likely to win the next trick as well. Playing Q will lose this trick, although A will win sometime in the future, the agent basically throws away a trick. ISSN correctly suggests so, assigning only 0.2% probability to this situation. In the second simulation the previous player

Table 3: Ablation studies in the number of layers and hidden neurons of the fully connected layer. Results are recorded in the metric of average trick loss.

| Network | Layers | Hidden neurons | Cost |
|---------|--------|----------------|-------|
| ISSN | 4 | 400 | 0.067 |
| ISSN | 4 | 800 | 0.066 |
| ISSN | 8 | 400 | 0.066 |
| ISSN | 8 | 800 | 0.064 |
| SSN | 4 | 400 | 0.027 |
| SSN | 4 | 800 | 0.026 |
| SSN | 8 | 400 | 0.025 |
| SSN | 8 | 800 | 0.024 |

Table 4: Ablation studies in the effect of data augmentation. Results are recorded in the metric of average trick loss.

| Network | Data Augmentation | Cost |
|---------|-------------------|-------|
| ISSN | No | 0.075 |
| ISSN | Yes | 0.064 |
| SSN | No | 0.052 |
| SSN | Yes | 0.024 |

Table 5: Ablation studies in the effect of the training target. Results are recorded in the metric of average trick loss.

| Training target | Cost |
|-----------------|-------|
| Simulation optimal moves counter | 0.041 |
| Indicator function of simulated best moves | 0.024 |

has ♠QT. Since ♠J is played, ♠QT are equal cards, and we expect a similar probability to both actions. ISSN suggests playing ♠Q from ♠QT is 51%. In the last simulation the previous player has only a single spade remaining, so he has to play that card. After these reasoning, the simulations are reweighed, so the agent focuses more on more likely simulations.

## 7    ABLATION STUDIES

We conduct the following ablation studies during the training of ISSN and SSN. First, we examine the effects of changing the number of layers and hidden neurons of the fully connected layers. The result is shown in Table 3. We can see that with more layers and more hidden units the performance increases, but the gain is minimal.

We find that data augmentation plays a critical role in the pretraining of the networks. The result is shown in Table 4. Without data augmentation, the performance drops significantly.

For the perfect information network to imitate DDS results, we can either choose the DDS simulation optimal move counter as the training target, or train on an indicator function of the simulated best moves only. The latter results in a sharper distribution, and performs better in the given setting. The result comparison is shown in Table 5.

## 8    CONCLUSION AND FUTURE WORK

In this work, we present Incomplete State Solver Network that can suggest actions with imperfect information. We use the network outputs to calculate the probabilities of simulations and reweigh them correspondingly. The weighted simulations perform better, and we demonstrate it in contract bridge playing phase. ISSN improves the performances of both ground truth simulations using DDS and network simulations with SSN. This approach is not specific to the contract bridge game, but general to all imperfect information games. We are also interested in long range belief modeling with recurrent networks for imperfect information games, and we leave this as future work.

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
