# OpenReview forum: "All Simulations Are Not Equal: Simulation Reweighing for Imperfect Information Games"
_ICLR.cc/2020/Conference — Reject_

### Official Review · AnonReviewer3 · 2019-10-21
**Official Blind Review #3**

**Rating:** 3

**Review:**



This paper deals imperfect information games, and builds a Bayesian method to model the unknown part of the current state, making use of the past moves (which constrain the game, here Contract Bridge).
This new Bayesian method is compared to Monte Carlo style techniques, which are much more computationally expensive (they draw random samples of the unknown part of the information and then solve the perfect-information version of the game, for each simulated possible full-state).
The work also introduces a Neural Network (NN) to estimate the best moves in the perfect-information version of the game (instead of making the full tree search of the optimal moves to play).
The final model proposed (ISSN+SSN) uses a NN combined with Bayesian computation, using the NN at each time step of the past to update the belief about the current missing information.

Overall the paper is written clearly: as a non-expert in RL, I was able to follow rather easily what is done.


However, the results are not convincingly good.  Maybe it is just the interpretation/contextualization that is insufficient.  I have 3 important remarks:

1. It is stated in the abstract "that it [the new method] outperforms previous state-of-the-art Monte Carlo simulation based methods, and achieves better play per decision."  However, the costs in the 'reweighing DDS' section (table 2) are lower (better) than those of the 'reweighing SSN' section.  The results show improvement upon using some reweighing, as ISSN+sthg is better than without ISSN (although by a very small relative decrease in cost).
1.a. I understand that the NN-based methods (last three lines of table 2) are incomparably faster than the DDS-based approaches (baseline). And the total rate of tricks loss per initial state remains low (6% seems small as an absolute value).  But, they are almost double of the DDS-based loss rates !  It is not clear how this can be considered 'outperforming' or 'better play per decision'.  If you can explain in which sense those are good results for the SSN vs DDS, please do so. If they are not, please do acknowledge it, and eventually contextualize (maybe a 2-fold increase of loss rate is okay given the large speedup obtained?).
1.b. in the same way, it is not clear how ISSN outperforms its no-memory counterparts, given the loss decreases are essentially negligible. Maybe one needs to increase the value of T to make ISSN's success more obvious?
2. In addition, why not compare this Bayesian method with other Bayesian methods (quoted at the very end of section 3)?  Here the paper focuses on comparison with 'deterministic'  methods, i.e. methods which sample complete states (simulations) to then solve the complete-information version of the game (via exhaustive tree search).  Those are what I would call brute-force methods.
However, although simulations may be done (expensively) for contract bridge, in some other cases this kind of sampling may become prohibitively expensive, so that only Bayesian methods are left.  If this kind of argument is the justification for your work, please make it more explicitly.  Otherwise please correct me and explain the context (the role of Bayesian methods, what has been done and what hasn't been) more clearly.
If you are to situate the work within the Bayesian-based approaches, the question remains: is ISSN+SSN better than other Bayesian-inspired, "information-completing" methods ?
To summarize, the paper convincingly shows that Bayesian-NN methods can compete with expensive brute force methods, but it would be very nice to see how the method introduced compares with other recent Bayesian approaches. (Or if no such comparison can be done, explain why).

3. In addition, here it seems your baseline is based on GIB, which was winning tournaments ~20 years ago: why not compare with Wbridge5 or JackBridge ?  I think you need to at least explain your choice.

Because of these weak points in terms of experimental results, I lean to reject the paper.  However, depending on the authors answers and clarifications on my 3 important remarks above, I am ready to change my rating.



Also, I have a couple of more or less minor remarks to improve the paper:
The definition of the cost is not explicitly given: "We set up the evaluation metric by tricks loss per deal. The ground truth play is DDS result for the original deal and all simulation deals. We compare all the following algorithmS with the ground truth to get costs of the policy."
You should rephrase this to make the definition of 'cost' very explicit. Is it simply the average rate of lost tricks (per given set of 13 cards in the agent hands) ?
This is quite crucial and make the reading of results a bit complicated (especially since results fo not match the conclusion announced in the abstract).

"DDS) 3 computes the maximum tricks  each side can get if all the plays are optimal"
In this place and a couple others, you should explicitly recall whether you mean 'assuming perfect information', or not.  Sometimes it can get confusing, and a bit of repetition won't hurt.  I think I understood correctly that DDS solves (perfectly) the perfect information game, but at times I thought other methods also made use of the full information (?)
Figure 4c is nicely explained and this part really illustrates well the idea of the method, I like it. Although, the notation AQ3 was not obvious for me at first, and I think it is worth improving this figure, making use of the right-hand-side space, to make it an autonomously explanatory figure.
'position': the term is not defined. I would guess it means the current state of the game (the agent's hand and the cards played in the past or during the current trick). This should be said explicitly.
Also "hands" seem to refer to the 4 hands (1 for each player) (is that correct?)

There are wrong singular/plurals ('s'/no 's') in several places. This is simple to correct and should be corrected.

in section 7, ablation studies. This is a very nice study, but you should precise how much you augment the data here (or recall by how much, if you say it earlier).

This passage is unclear and should be rephrased for clarity:
"We run DDS on all 2.4 million positions, to get the
corresponding results for each available action. This serves as ground truth of the current position.
For each position, we run 50 simulations to compute the baselines and serve as training target of the
pretrained network. Each call to DDS takes about 5 milliseconds."

This passage is unclear and should be rephrased for clarity:
"For simplicity, in this work we just use discarding information."

This passage is slightly unclear and should be rephrased for clarity:
"For each simulation, the moves with the optimal number of
tricks are marked with optimal moves. We sum up the optimal moves counter in these k simulations.
This results in a counter for each legal move and we treat this as the training target. The process is
described in Figure 1.""


**Experience Assessment:**

I do not know much about this area.

**Review Assessment: Checking Correctness Of Derivations And Theory:**

I assessed the sensibility of the derivations and theory.

**Review Assessment: Checking Correctness Of Experiments:**

I assessed the sensibility of the experiments.

**Review Assessment: Thoroughness In Paper Reading:**

I read the paper thoroughly.

---

> ### Author Response · Authors · 2019-11-12
> **Reply to R3**
>
> We thank the reviewers for the insightful feedbacks.
> First, We sincerely apologize the grammatical errors and will make a revision to correct all of them.
>
> Explanation of reweighing results:
> Q1 The understanding of ISSN reweighing is correct, the performance improves. The rows across “Reweighing DDS” and “Reweighing SSN” are not meant to be directly compared. Since the cost is small enough, we are trading accuracy for speed, and with a very tight computational budget we are still able to get decent results. We are doing more analysis on which situations the method performs better.
>
> Q2 We have tried to use Bayesian Action Decoder as a baseline method. However BAD requires to track the whole possible state space (millions in the beginning of Hanabi, but quickly reduces to hundreds ~ thousands, because cards are played and hinted). However, during the bidding phase the number of card combinations for an agent is C(52, 13), and does not decrease. This makes BAD intractable, and is one of the reasons that we choose to use simulation based Bayesian method. We have also tested BAD on a mini-version of bridge such that each player only holds 2 suits and 5 cards. We enumerate all the possible states and make BAD belief updates, but it cannot converge to the optimal solution.  To the best of the author’s knowledge there is no other similar baseline work done for imperfect information games.
>
> Q3 To the best of the author’s knowledge GIB, Jack and Wbridge5 all use a similar engine for the playing phase, except for the human designed heuristic part. Some of the heuristics include information gathered from the bidding phase. Also, Jack and Wbridge5 are close sourced commercial software so we are not sure what heuristics are used exactly. Thus we compared against the simplest baseline.
>
> Other:
> Yes, all the cost is average trick loss from optimal play, and optimal play is always assumed perfect information
>
> Why playing “Q” from AQ3 is unusual: When it goes J-K to a player, if you play A you will win the current trick, and since AK are already played, Q is very likely to win the next trick as well (if it is not trumped). Playing Q will lose this trick, although A will win sometime in the future, the agent basically throws away a trick. That’s why playing Q with AQ3 is very unlikely. We will put this explanation paragraph in the next revision.
>
> When “hands” are referring to all 4 hands, we are switching to call it “deal”
>
> Data augmentation: we stated in the main text (end of 4.4.1) that we augment the data by shuffling suit.

---

### Official Review · AnonReviewer2 · 2019-10-22
**Official Blind Review #2**

**Rating:** 3

**Review:**

This paper presents an approach to playing imperfect information games, an “Incomplete State Solver Network” (ISSN) within the domain of contact bridge. The paper’s primary technical contributions are the network, and a large dataset of contact bridge games, which the authors make publicly available.

I believe that the work that the authors did in regards to this paper is valuable. However, I had a very hard time following along with the paper. The major issue is the language, which I mean both in terms of particular phrases or terms going unexplained, and the grammar and phrasing of the sentences.

The most clear early example of terms and phrases going unexplained is the second section describing contract bridge. Many terms are used without any explanation. For example, what a target contract is or what a trump means in the case of contract bridge. This made the remainder of the paper, including the results difficult to understand.

As an indication of the grammar and phrasing issues in the paper I have below included issues from just the first page of the paper:
- “In real world”-> “In the real world”
- “have to made decision” -> “have to make decisions”
- “researchers steers towards” -> “researchers have focused on”
- “Chess, best action” -> “Chess, the best action”
- “it is independent of opponent and action history” -> “it is independent of the opponent or action history”
- “history actions” -> “action histories”
- “In early ages there are heuristic based system to assign different prior” -> “Early approaches employ heuristic based systems to assign different priors”
- “try to convert the problem to a perfect” -> “try to convert the problem into a perfect”
- “explicitly with neural networks, and try to optimize it” -> “explicitly with neural networks, trying”
- (not a phrasing issue but I would have appreciated a citation for the claim at the end of the third paragraph of the intro)
- “Simulation based approach usually requires large” -> “Simulation based approaches usually require a large”
- “sequences of existing player and opponent” -> “sequences of the existing player and opponent”
- “a few underlying complete information state” -> “a few underlying complete information states”

This issue of readability came up in the figures as well. The figures in the paper are dense and very difficult to parse. There is little explanation in the text, and I found them difficult to glean anything from without an understanding of contact bridge.

From what I can gather from the paper this is good and valuable work. However, I think the paper is not yet ready for publication as a communication of this work.


**Experience Assessment:**

I have read many papers in this area.

**Review Assessment: Checking Correctness Of Derivations And Theory:**

I did not assess the derivations or theory.

**Review Assessment: Checking Correctness Of Experiments:**

I assessed the sensibility of the experiments.

**Review Assessment: Thoroughness In Paper Reading:**

I made a quick assessment of this paper.

---

> ### Author Response · Authors · 2019-11-12
> **Reply to R2**
>
> We thank the reviewers for the insightful feedbacks.
> First, We sincerely apologize the grammatical errors and will make a revision to correct all of them.
>
> We list the number of confused terms here:
>
> 1. Target contract (Explained in Section 2.2): number of tricks (winning rounds) required to make the contract. If the team makes the contract then receives positive rewards (if the contract is higher, e.g., 7spade, then the reward is higher), otherwise receives negative rewards.
> 2. Trump suit/card (Explained in Section 2.2): The suit/card that could be used to beat cards in a normal suit where no cards is available.
>
> There might be some unclarity in the original text and we are working on rephrasing it.
>
> For the figures:
> We do not expect readers to read into specific hand and cards, but show an overall idea of the training process. More detailed explanation for the figure are in the main text, for example, how ISSN is trained and what is the process of simulation reweighing.

---

### Official Review · AnonReviewer4 · 2019-11-12
**Official Blind Review #4**

**Rating:** 1

**Review:**

This paper introduces two networks that are trained to predict DDS. While one is trained with perfect information, the other one (ISSN) with imperfect information.
The ISSN is then used to compute posterior probability distribution (based on a history leading to the current state). The ideas is that such posterior distribution should perform better compared to uniform distribution when used in determinization process.

I like the idea / motivation of the paper, but the authors could do a better job of explaining the motivation to people less familiar with techniques based on the determinization framework.
I also like the baselines that they chose to compare against - but the resulting comparison is far from perfect (see Issues section).

Minor issues:
 -  Please do a careful language check - the grammar is wrong in many places (most notably plural/singular nouns).
While this does not hurt the semantics, it makes it sometimes cumbersome to read.

  - Since this work is mostly about using non-uniform distribution during the determinization process, I think it's worthwhile to also mention [Whitehouse, Daniel. Monte carlo tree search for games with hidden information and uncertainty. Diss. University of York, 2014.] as reference point.

Issues:
 - My biggest issue is the experimental and evaluation section. The reported improvements seem small, but most importantly - it is impossible to asses the relevance of the results.
  There are no confidence intervals or variance reported. Given the seemingly small improvements, this could easily be noise?

 - While I am not certain, I assume that your numbers in Table 2 come from the 'test' split of the data - one would guess you used that split to stop the training (select the best model)?
  If that is the case, I don't think you can use the same split during the evaluation (even though you evaluate differently) - the reported numbers will be biased.

Improvement Suggestions
  - Please see my issues with the evaluation.
  - You say you will release the data and code - that is great, do it!
  - Your figures are way too large for what they do. I think you should make them much more compact and use the resulting space to improve and expand the experimental section. Please add lot more details about the evaluation.

Summary:
I think the paper is looking into an interesting problem and is going in the right direction, but the experimental section is at this point no good enough to suggest an acceptance.

**Experience Assessment:**

I have published in this field for several years.

**Review Assessment: Checking Correctness Of Derivations And Theory:**

I assessed the sensibility of the derivations and theory.

**Review Assessment: Checking Correctness Of Experiments:**

I carefully checked the experiments.

**Review Assessment: Thoroughness In Paper Reading:**

I read the paper thoroughly.

---

### Author Response · Authors · 2019-11-15
**Paper revision**

Thanks the reviewer for the insightful feedbacks.

We sincerely apologize for the grammar errors and have updated a revision to correct them. We have also cited the mentioned work in the new revision. For the experiments we use trick losses compared with optimal play assuming perfect information as the evaluation metric.

Here is a summarization of our contributions.

Existing methods, such as Counterfactual Regret Minimization, can handle incomplete information games with small belief space, e.g. Two-player Poker [1] (51*52/2 = 1326 possible card configuration), Hanabi [2] (only your own cards are not visible. #hidden states is around 10 millions and quickly drops when more cards are played) and Avalon the Resistance [3] (belief space is 60 for 5 players with different hidden roles). In contrast, in Contract Bridge, the hidden information space is large, since 2 out of 4 players' cards are unknown, which is about C(26, ``13) = 10 millions possibilities and unlike Hanabi, this number will not drop quickly over playing.

In this paper we want to address this issue by handling exponentially large hidden information space in Contract Bridge via a hybrid approach of sampling and neural network.  We show that a pre-trained neural network on millions of data with ground truth score obtained by complete information (from DDS solver) can predict the next best action given incomplete information fairly accurately, without intensive belief sampling. Furthermore, by reweighing the samples using past action history, we show the performance can be further improved, compared to multiple baselines. In summary, with a tight computation budget we can reach decent performance without a single call to the expensive DDS solver. We also perform a number of ablation studies.

[1] Superhuman AI for heads-up no-limit poker: Libratus beats top professionals, N. Brown and T. Sandholm. Science, 2017.

[2] The Hanabi Challenge: A New Frontier for AI Research, N. Bard et al, arXiv 2018

[3] Finding Friend and Foe in Multi-Agent Games, J. Serrino et al, NeurIPS 2019

---

### Decision · Program_Chairs · 2019-12-19

**Decision:**

Reject

**Comment:**

A method is introduced to estimate the hidden state in imperfect information in multiplayer games, in particular Bridge. This is interesting, but the paper falls short in various ways. Several reviewers complained about the readability of the paper, and also about the quality and presentation of the interesting results.

It seems that this paper represents an interesting idea, but is not yet ready for publication.